# Neural Shape Deformation Priors

**Jiapeng Tang**[1]    **Lev Markhasin**[2]    **Bi Wang**[2]    **Justus Thies**[3]    **Matthias Nießner**[1]

[1] Technical University of Munich    [2] Sony Europe RDC Stuttgart
[3] Max Planck Institute for Intelligent Systems, Tübingen, Germany

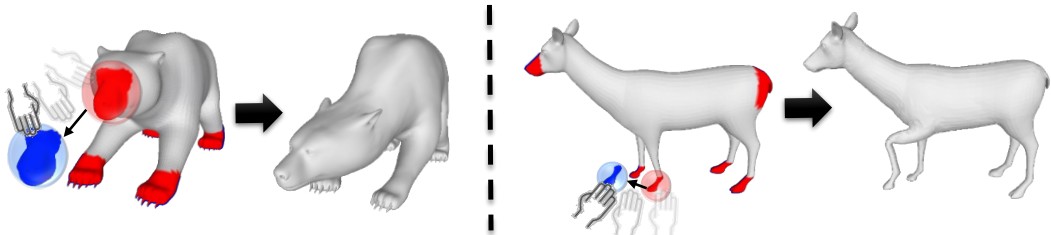

Figure 1: Neural shape deformation priors allow for intuitive shape manipulation of existing source meshes. A user can create novel shapes by dragging handles (red circles) defined on the region of interest (red regions) to desired locations (blue circles).

## Abstract

We present *Neural Shape Deformation Priors*, a novel method for shape manipulation that predicts mesh deformations of non-rigid objects from user-provided handle movements. State-of-the-art methods cast this problem as an optimization task, where the input source mesh is iteratively deformed to minimize an objective function according to hand-crafted regularizers such as ARAP [1]. In this work, we learn the deformation behavior based on the underlying geometric properties of a shape, while leveraging a large-scale dataset containing a diverse set of non-rigid deformations. Specifically, given a source mesh and desired target locations of handles that describe the partial surface deformation, we predict a continuous deformation field that is defined in 3D space to describe the space deformation. To this end, we introduce transformer-based deformation networks that represent a shape deformation as a composition of local surface deformations. It learns a set of local latent codes anchored in 3D space, from which we can learn a set of continuous deformation functions for local surfaces. Our method can be applied to challenging deformations and generalizes well to unseen deformations. We validate our approach in experiments using the DeformingThing4D dataset, and compare to both classic optimization-based and recent neural network-based methods.

## 1 Introduction

Editing and deforming 3D shapes is a key component in animation creation and computer aided design pipelines. Given as little user input as possible, the goal is to create new deformed instances of the original 3D shape which look natural and behave like real objects or animals. The user input is assumed to be very sparse, such as vertex handles that can be dragged around. For example, users can animate a 3D model of an animal by dragging its feet forward. This problem is severely ill-posed and typically under-constrained, as there are many possible deformations that can be matched with the provided partial surface deformations of handles, especially for large surface deformations. Thus, strong priors encoding deformation regularity are necessary to tackle this problem. Physics and

36th Conference on Neural Information Processing Systems (NeurIPS 2022).

differential geometry provide solutions that use various analytical priors which define natural-looking mesh deformations, such as elasticity [2, 3], Laplacian smoothness [4, 5, 6], and rigidity [1, 7, 8] priors. They update mesh vertex coordinates by iteratively optimizing energy functions that satisfy constraints from both the pre-defined deformation priors and given handle locations. Although these algorithms can preserve geometric details of the original source model, they still have limited capacity to model realistic deformations, since the deformation priors are region independent, e.g., the head region deforms in a similar way as the tail of an animal, resulting in unrealistic deformation states.

Hence, motivated by the recent success of deep neural networks for 3D shape modeling [9, 10, 11, 12, 13, 14, 15, 16, 17, 18, 19, 20, 21, 22], we propose to learn shape deformation priors of a specific object class, e.g., quadruped animals, to complete surface deformations beyond observed handles. We formulate the following properties of such a learned model; (1) it should be robust to different mesh quality and number of vertices, (2) the source mesh is not limited to canonical pose (i.e., the input mesh can have arbitrary pose), and (3) it should generalize well to new deformations. Towards these goals, we represent deformations as a continuous deformation field which is defined in the near-surface region to describe the space deformation caused by the corresponding surface deformation. The continuity property enables us to manipulate meshes with infinite number of vertices and disconnected components. To handle source meshes in arbitrary poses, we learn shape deformations via canonicalization. Specifically, the overall deformation process consists of two stages: arbitrary-to-canonical transformation and canonical-to-arbitrary transformation. To obtain more detailed surface deformations and better generalization capabilities to unseen deformations, we propose to learn local deformation fields conditioned on local latent codes encoding geometry-dependent deformation priors, instead of global deformation fields conditioned on a single latent code. To this end, we propose Transformer-based Deformation Networks (TD-Nets), which learns encoder-based local deformation fields on point cloud approximations of the input mesh. Concretely, TD-Nets encode an input point cloud with surface geometry information and incomplete deformation flow into a sparse set of local latent codes and a global feature vector by using the vector attention blocks proposed in [23]. The deformation vectors of spatial points are estimated by an attentive decoder, which aggregates the information of neighboring local latent codes of a spatial point based on the feature similarity relationships. The aggregated feature vectors are finally passed to a multi-layer-perceptron (MLP) to predict displacement vectors which can be applied to the source mesh to compute the final output mesh.

To summarize, we introduce transformer-based local deformation field networks which are capable to learn shape deformation priors for the task of user-driven shape manipulation. The deformation networks learn a set of anchor features based on a vector attention mechanism, enhancing the global deformation context, and selecting the most informative local deformation descriptors for displacement vector estimations, leading to an improved generalization ability to new deformations. In comparison to classical hand-crafted deformation priors as well as recent neural network-based deformation predictors, our method achieves more accurate and natural shape deformations.

## 2    Related Work

User-guided shape manipulation lies at the intersection of computer graphics and computer vision. Our proposed method is related to polygonal mesh geometry processing, neural field representations, as well as vision transformers.

**Optimization-based Shape Manipulation.**    Classical methods formulate shape manipulation as a mathematical optimization problem. They perform mesh deformations by either deforming the vertices [24, 25] or the 3D space [26, 27, 8, 28, 29]. Performing mesh deformation without any other information about the target shape, but only using limited user-provided correspondences is an under-constrained problem. To this end, the optimization methods require deformation priors to constraint the deformation regularity as well as the smoothness of the deformed surface. Various analytic priors have been proposed which encourage smooth surface deformations, such as elasticity [2, 3], Laplacian smoothness [4, 5, 6], and rigidity [1, 7, 8]. These methods use efficient linear solvers to iteratively optimize energy functions that satisfy constraints from both the pre-defined deformation prior and provided handle movements. Recently, NFGP [30] was proposed to optimize neural networks with non-linear deformation regularizations. Specifically, it performs shape deformations by warping the neural implicit fields of the source model through a deformation vector field, which is constrained by

modeling implicitly represented surfaces as elastic shells. NeuralMLS [31] learned a geometry-aware weight function of a shape and given control points for moving least squares(MLS) deformations, which smoothly interpolates the control point displacements over space. Although they can preserve many geometric details of the source shape, they struggle to model complex deformations, as local surfaces are simply constrained to be transformed in a similar manner. In contrast, we aim to learn deformation priors based on local geometries to infer hidden surface deformations.

**Learning-based Shape Reconstruction and Manipulation.** Learning-based shape manipulation has been studied to learn shape priors based on shape auto-encoding or auto-decoding. [32, 33, 34, 35] map a class of shapes into a latent space. During inference, given handle positions as input, they find an optimal latent code whose 3D interpretation is the most similar to the observation. In contrast, we learn explicit deformation priors to directly predict 3D surface deformations. Jakab et al. [36] proposed to control shapes via unsupervised 3D keypoint discovery. Instead, we use partial surface deformations represented by handle displacements as input observations, rather than keypoint displacements. There exist a series of methods that use deep neural networks to complete non-rigid shapes [35, 37, 38, 39, 40, 41, 42, 43] from partial scans. Our task is partially related to this task, but our shape manipulation task from user input requires completion of the deformation field. In contrast to shape completion, our setting is more under-constrained, as the user-provided handle correspondences are very sparse and more incomplete than partial point clouds from scans. Recent methods for clothed-human body reconstruction choose to canonicalize the captured scan into a pre-defined T-pose [44, 45, 46] using the skeletal deformation model of SMPL [47] or STAR [48] which can also be used to later animate the human. Inspired by this, we also perform a canonicalization to enable editing of source meshes with arbitrary poses, before applying the actual deformation towards the target pose handles.

**Continuous Neural Fields.** Continuous neural field representations have been widely used in 3D shape modeling [9, 11, 10] and 4D dynamics capture [49, 40, 38, 37, 39]. Recent work that represents 3D shapes as continuous signed distance fields [17, 12, 18, 19, 20] or occupancy fields [9, 11, 14, 50, 15, 16, 51, 21, 52, 53] can theoretically obtain volumetric reconstructions with infinite resolutions, as they are not bound to the resolution of a discrete grid structure. Similarly, we learn continuous deformation fields defined in 3D space for shape deformations [13, 35, 30, 54]. Due to the continuity of the deformation fields, our method is not limited by the number of mesh vertices, or disconnected components. Different from ShapeFlow [35], OFlow [49], LPDC-Net [40] and NPMs [37] that learn a deformation field from a single latent code, inspired by local implicit field learning [14, 15, 21, 52, 55], we model the deformation field as a composition of local deformation functions, improving the representation capability of describing complex deformations as well as generalization to new deformations.

**Visual Transformers.** Recently, transformer architectures [56] from natural language processing have revolutionized many computer vision tasks, including image classification [57, 58], object recognition [59], semantic segmentation [60], or 3D reconstruction [61, 62, 52, 55, 63]. We refer the reader to [64] for a detailed survey of visual transformers. In this work, we propose the usage of a transformer architecture to learn deformation fields. Given the input point cloud sampled from the source mesh with partial deformation flow (defined by the user handles), we employ the vector attention blocks from Point Transformer [23] as a main point cloud processing module to extract a sparse set of local latent codes, enhancing the global understanding of deformation behaviours. Based on the obtained local deformation descriptors, our attentive deformation decoder learns to attend to the most informative features from near-by local codes to predict a deformation field.

## 3  Approach

Given a source mesh $\mathcal{S} = \{\mathcal{V}, \mathcal{F}\}$ where $\mathcal{V}$ and $\mathcal{F}$ denote the set of vertices and the set of faces, respectively, we aim to deform $\mathcal{S}$ to obtain a target mesh $\mathcal{T}$ by selecting a sparse set of mesh vertices $\mathcal{H} = \{\mathbf{h}_i\}_{i=1}^{\ell}$ as handles, and dragging them to target locations $\mathcal{O} = \{\mathbf{o}_i\}_{i=1}^{\ell}$. The key idea in this work is to use deformation priors to complete hidden surface deformations. Specifically, the goal is to learn a continuous deformation field $\mathbf{D}$ defined in 3D space, from which we can obtain the deformed mesh $\mathcal{T}' = \{\mathcal{V} + \mathbf{D}(\mathcal{V}), \mathcal{F}\}$ through vertex deformations of the source mesh $\mathcal{S}$. The overall pipeline of the proposed approach is shown in Figure 2. Our method can be applied to input meshes

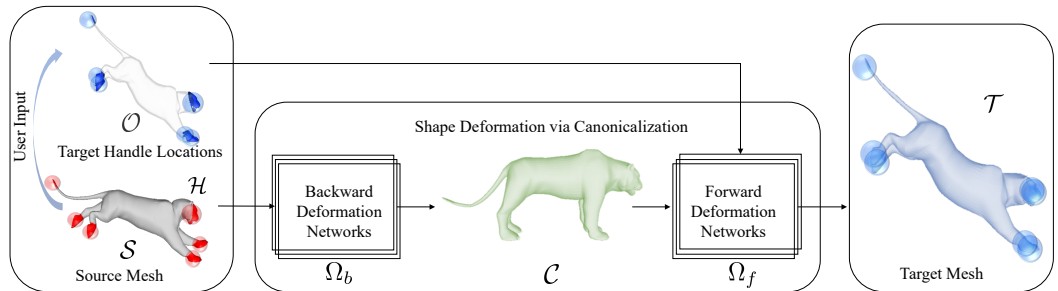

Figure 2: **Overview**. Given a source mesh $\mathcal{S}$ with sparse handles $\mathcal{H}$ (red circles) and their respective target locations $\mathcal{O}$ (blue circles) as input, our method deforms the mesh to the target mesh $\mathcal{T}$ via canonicalization $\mathcal{C}$. The backward $\Omega_b$ and forward $\Omega_f$ deformation networks store the deformation priors that allow our method to produce consistent and natural-looking outputs.

in arbitrary poses by leveraging learned shape deformation via canonicalization (see Section 3.1). To represent the underlying deformation prior, we propose neural deformation fields as described in Section 3.2 which can be learned from large deformation datasets (see Section 3.3).

## 3.1 Learning Shape Deformations via Canonicalization

To ensure robustness w.r.t. varying input mesh quality (topology and resolution), we operate on point clouds instead of meshes. Specifically, we sample a point cloud $\mathcal{P}_{\mathcal{S}} = \{\mathbf{p}_i\}_{i=0}^{n} \in \mathbb{R}^{n \times 3}$ from $\mathcal{S}$ of size $n = 5000$. We define the target handle point locations $\mathcal{P}_{\mathcal{O}} = \{\mathbf{o}_i\}_{i=0}^{n} \in \mathbb{R}^{n \times 3}$, where we use zeros to represent unknown point flows. Further, to avoid the ambiguity of zero point flow, we define the corresponding binary user handle masks $\mathcal{M} = \{b_i\}_{i=0}^{n} \in \mathbb{R}^n$ where $b_i = 1$ if $\mathbf{p}_i$ is a handle or otherwise $b_i = 0$.

To learn the shape transformation between two arbitrary non-rigidly deformed poses, one can learn deformation fields that directly map the source deformed space to target space. However, it would be difficult to learn the deformation priors well, as there could be infinite deformation state transformation pairs. To decrease the learning complexity, we introduce a canonical space as an intermediate state. We divide the shape transformation process into two steps; a backward deformation that aligns the source deformed space to canonical space, and a forward deformation that maps the canonical space to the target deformation space. Concretely, $\mathcal{P}_{\mathcal{S}}$ is passed into the backward transformation network $\Omega_b$ to learn the backward deformation field $\mathbf{D}_b$ which transforms the input shape $\mathcal{P}_{\mathcal{S}}$ into a canonical pose $\mathcal{P}'_{\mathcal{C}}$. Similarly, the querying non-surface point set $\mathcal{Q}_{\mathcal{S}} = \{\mathbf{q}_i\}_{i=0}^{m} \in \mathbb{R}^{m \times 3}, m = 5000$ randomly sampled in the 3D space of $\mathcal{S}$ is also mapped to canonical space through $\mathcal{Q}'_{\mathcal{C}} = \mathcal{Q}_{\mathcal{S}} + \mathbf{D}_b(\mathcal{Q}_{\mathcal{S}})$. Lastly, given $\mathcal{P}'_{\mathcal{C}}$, $\mathcal{M}$, and $\mathcal{P}_{\mathcal{O}}$ as input, a forward transformation network $\Omega_f$ is learned to represent the forward deformation field $\mathbf{D}_f$ that predicts final locations $\mathcal{Q}'_{\mathcal{T}} = \mathcal{Q}'_{\mathcal{C}} + \mathbf{D}_f(\mathcal{Q}'_{\mathcal{C}})$.

## 3.2 Transformer-based Deformation Networks (TD-Nets)

The deformation via canonicalization is based on two deformation field predictors (forward and backward deformations). Both networks share the same architecture, thus, in the following, we will only describe the forward deformation network as visualized in Figure 3 while the backward deformation network is analogous. It consists of a transformer-based deformation encoder and a vector cross attention-based decoder network.

**Point transformer encoder.** Given a point set $\mathcal{P}_{\mathcal{C}}$ with handle locations $\mathcal{P}_{\mathcal{O}}$ and a binary mask $\mathcal{M}$ as inputs, we use point transformer layers from [23] to build our encoder modules. The point transformer layer is based on the vector attention mechanism [65]. Let $\mathcal{X} = \{\mathbf{x}_i, \mathbf{f}_i\}_i$ and $\mathcal{Y} = \{\mathbf{y}_i, \mathbf{g}_i\}_i$ be the query and key-value sequences, where $\mathbf{x}_i$ and $\mathbf{y}_i$ denote the coordinates of query and key-value points with corresponding feature vectors $\mathbf{f}_i$ and $\mathbf{g}_i$. The vector cross attention operator VCA is defined as:

$$\text{VCA}(\mathcal{X}, \mathcal{Y}): \quad \mathbf{f}'_i = \sum_{j \in \mathcal{N}_i} \rho(\gamma(\varphi(\mathbf{g}_j) - \psi(\mathbf{f}_i) + \delta)) \odot (\alpha(\mathbf{f}_i) + \delta), \tag{1}$$

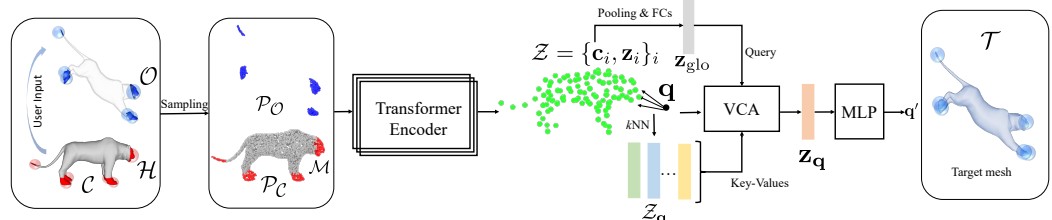

Figure 3: **Transformer-based Forward Deformation Networks**. Given a canonical mesh $\mathcal{C}$ with handle positions $\mathcal{H}$ (red circles) and desired handle locations $\mathcal{O}$ (blue circles), we perform surface sampling to obtain a point cloud $\mathcal{P}_\mathcal{C}$ with additional channels of handle mask $\mathcal{M}$ and point flow $\mathcal{P}_\mathcal{O}$. A point-transformer encoder is devised to extract a sparse set of local latent codes $\mathcal{Z} = \{\mathbf{c}_i, \mathbf{z}_i\}_i$ from this point cloud, where $\mathbf{c}_i$ are the anchor positions of the latent features $\mathbf{z}_i$. For a specific point $\mathbf{q}$ in 3D space (i.e. a vertex from the source mesh), based on the $\mathbf{z}_{\mathrm{glo}}$, a vector cross attention (VCA) block is used to effectively fuse the information of $\mathcal{Z}_\mathbf{q}$ into $\mathbf{z}_\mathbf{q}$ from the $k$ nearest neighbouring latent codes of $\mathbf{q}$. Using a multi-layer perceptron (MLP) conditioned on $\mathbf{z}_\mathbf{q}$, we predict the deformed location $\mathbf{q}'$ in the target space.

where $\mathbf{f}'_i$ are the aggregated features, $\varphi$, $\psi$, and $\alpha$ are linear projections implemented by a fully-connected layer. $\gamma$ is a mapping function implemented by a two-layer MLP to predict attention vectors. $\rho$ is the attention weight normalization function, in our case *softmax*. $\delta := \theta(\mathbf{x}_i - \mathbf{y}_j)$ is the positional embedding module [56, 66] implemented by a two linear layers with a single ReLU [67]. It leverages relatively positional information of $\mathbf{x}_i$ and $\mathbf{y}_j$ to benefit the network training. Then, with the definition of VCA, the vector self-attention operator VSA can be defined as:

$$\mathrm{VSA}(\mathcal{X}) := \mathrm{VCA}(\mathcal{X}, \mathcal{X}). \tag{2}$$

Based on VCA and VSA, we can define two basic modules to build our encoder network, i.e. the *point transformer block* (PTB) and the *point abstraction block* (PAB). The definition of the point transformer block PTB is a combination of the BatchNorm (BN) layer [68], VSA, and residual connections, formulated as:

$$\mathrm{PTB}(\mathcal{X}) := \mathrm{BN}(\mathcal{X} + \mathrm{VSA}(\mathcal{X})). \tag{3}$$

For each point $\mathcal{X}_i$, it encapsulates the information from $k_{\mathrm{enc}} = 16$ nearest neighborhoods while keeping the point's position $\mathbf{x}_i$ unchanged. The point abstraction block PAB consists of farthest point sampling (FPS), BN, VCA, and VSA, which is defined as follow:

$$\mathrm{PAB}(\mathcal{X}) := \mathrm{BN}(\mathrm{FPS}(\mathcal{X}) + \mathrm{VSA}(\mathrm{VCA}(\mathrm{FPS}(\mathcal{X}), \mathcal{X}))). \tag{4}$$

The point cloud $\mathcal{P}_\mathcal{C}$ with handle mask $\mathcal{M}$ and flow $\mathcal{P}_\mathcal{O}$ as additional channels are passed to a point transformer block (PTB) to obtain a feature point cloud $\mathcal{Z}_0 = \{\mathbf{c}_i^0, \mathbf{z}_i^0\}_{i=1}^n$. By using two consecutive point abstraction blocks (PABs) with intermediate set size of $n_1 = 500$ and $n_2 = 100$, we obtain $\mathcal{Z}_1 = \{\mathbf{c}_i^1, \mathbf{z}_i^1\}_{i=1}^{n_1}$ and $\mathcal{Z}_2 = \{\mathbf{c}_i^2, \mathbf{z}_i^2\}_{i=1}^{n_2}$. To enhance global deformation priors, we stack 4 point transformer blocks with full self-attention whose $k_{\mathrm{enc}}$ is set to 100 to exchange the global information in the whole set of $\mathcal{Z}_2$. By doing so, we can obtain a sparse set of local deformation descriptors $\mathcal{Z} = \{\mathbf{c}_i, \mathbf{z}_i\}_{i=1}^{100}$ that are anchored in $\{\mathbf{c}_i\}$. Finally, we perform a global max-pooling operation followed by two linear layers to obtain the global latent vector $\mathbf{z}_{\mathrm{glo}}$.

**Attentive deformation decoder.** Based on the learned local latent codes $\mathcal{Z} = \{\mathbf{c}_i, \mathbf{z}_i\}_{i=1}^{100}$ and global latent vector $\mathbf{z}_{\mathrm{glo}}$, the deformation decoder defines the forward deformation function $\mathbf{D}_f : \mathbb{R}^3 \to \mathbb{R}^3$, which maps a point $\mathbf{q}$ from the canonical space of $\mathcal{C}$ to the 3D space of $\mathcal{T}$. Similar to tri-linear interpolation operations in grid-based implicit field learning, a straightforward way to find the corresponding feature vector $\mathbf{z}_\mathbf{q}$ is to use the weighted combination of $k_{\mathrm{dec}} = 16$ nearby local codes $\mathcal{Z}_\mathbf{q} = \{\mathbf{c}_k, \mathbf{z}_k\}_{k=1}^{k_{\mathrm{dec}}}$. Intuitively, the weight is inversely proportional to the euclidean distance between $\mathbf{q}$ and the anchoring location $\mathbf{c}_k$ [15]. However, distance-based feature queries ignore the relationships between deformation descriptors. Thus, we propose to obtain $\mathbf{z}_\mathbf{q}$ by adaptively aggregating information of $\mathcal{Z}_\mathbf{q}$ based on the vector cross-attention operator:

$$\mathbf{z}_\mathbf{q} = \mathrm{VCA}(\{\mathbf{q}, \mathbf{z}_{\mathrm{glo}}\}, \mathcal{Z}_\mathbf{q}). \tag{5}$$

The local information aggregation enables us to flexibly search the local deformation priors, thus, improving the generalizability to new deformations. Finally, the $\mathbf{z}_\mathbf{q}$ is fed into an MLP composed of five Res-FC blocks to estimate the associate location $\mathbf{q}' = \mathbf{q} + \mathbf{D}_f(\mathbf{q}; \mathbf{z}_\mathbf{q})$ in the target space.

## 3.3 Training Objectives

For training, we need a set of triplets $(\mathcal{S}, \mathcal{C}, \mathcal{T})$ with dense correspondences, from which we can randomly sample surface point clouds $(\mathcal{P}_{\mathcal{S}}, \mathcal{P}_{\mathcal{C}}, \mathcal{P}_{\mathcal{T}})$ of size $n$ and querying non-surface points $(\mathcal{Q}_{\mathcal{S}}, \mathcal{Q}_{\mathcal{C}}, \mathcal{Q}_{\mathcal{T}})$ of size $m$ in the 3D space. To optimize the backward deformation networks, we employ the mean $\ell_2$ distance error that measures the difference between deformed points from source space and their ground-truths in the canonical space:

$$L_b = ||\Omega_b(\mathcal{P}_{\mathcal{S}}) - \mathcal{P}_{\mathcal{C}}||_2^2 + ||\Omega_b(\mathcal{Q}_{\mathcal{S}}) - \mathcal{Q}_{\mathcal{C}}||_2^2. \tag{6}$$

Similarly, to optimize the forward deformation networks, we use the following loss function:

$$L_f = ||\Omega_f(\mathcal{P}_{\mathcal{C}}) - \mathcal{P}_{\mathcal{T}}||_2^2 + ||\Omega_b(\mathcal{Q}_{\mathcal{C}}) - \mathcal{Q}_{\mathcal{T}}||_2^2 \tag{7}$$

The total loss function for source-target shape deformations is defined as:

$$L_{\text{total}} = ||\Omega_f(\Omega_b(\mathcal{P}_{\mathcal{S}})) - \mathcal{P}_{\mathcal{T}}||_2^2 + ||\Omega_f(\Omega_b(\mathcal{Q}_{\mathcal{S}})) - \mathcal{Q}_{\mathcal{T}}||_2^2. \tag{8}$$

# 4 Experiments

**Dataset.** Our experiments are performed on the DeformingThing4D-Animals [39] dataset which contains 1494 non-rigidly deforming animations with various motions comprising 40 identities of 24 categories. For the train/test split, we divide all animations into training (1296) and test (198). Similar to the D-FAUST [69] used in OFlow [49], the test set is composed of two subsets: (S1) contains 143 sequences of new motions for seen train identities, and (S2) contains 55 sequences of unseen individuals (and thus also new motions). During training, we randomly sample two frames from an identity as source-target deformation pairs. During inference, we consider the first frame of an animation as source mesh, and other frames as target meshes. To evaluate the generalization ability to unseen identities, we evaluate the pre-trained models on the animal dataset used in Deformation Transfer [70]. For the quantitative comparison on each test subset, we compute evaluation metrics for 300 randomly sampled pairs. In addition, we also include comparisons on another animal dataset used in TOSCA [71]. TOSCA [71] does not have correspondences between different poses of the same animal, and hence does not easily provide handle displacements as input. Thus, we provide a qualitative comparison under the setting of using user-specified handles as inputs.

**Implementation details.** Our approach is built on the PyTorch library [72]. Please refer to the supplementary material for the details of our network architecture. Our model consists of two training stages. We use an Adam [73] optimizer with $\beta_1 = 0.9$, $\beta_2 = 0.999$, and $\epsilon = 10^{-8}$. In the first stage, we train the forward and backward deformation networks individually. Specifically, the backward and forward deformation networks are respectively optimized by the objective described in Equations 6 or 7 using a batch size of 16 with the learning rate of 5e-4 for 100 epochs. In the second stage, the whole model is trained according to Equation 8 in an end-to-end manner using a batch size of 6 with a learning rate of 5e-5 for 20 epochs.

**Baselines.** We conduct comparisons against classical optimization-based and recent neural network-based methods. For the former, we select a representative work, ARAP [1], that constrains each local surface to be rigidly transformed as much as possible. For the latter, we compare our method with the learning-based deformation predictor ShapeFlow [35] that embeds each shape into a latent space and learns flow-based deformations among 3D shapes. We also compare to NFGP [30], a deep optimization method, which constrains the implicitly represented surfaces as elastic shells during the deformation process.

**Evaluation metrics.** We consider $\ell_2$ distance error of mesh vertices ($\ell_2 \times 0.001$), Chamfer Distance (CD $\times 0.01$) of sampled point clouds of 30k points, and Face Normal Consistency (FNC $\times 0.01$) as primary evaluation metrics. Please refer to the supplementary material for a detailed explanation of these metrics. Note that for $\ell_2$ and CD, lower is better, while for FNC, higher is better.

## 4.1 Comparisons

For a qualitative comparison, we visualize the vertex $\ell_1$ distance error maps of deformed meshes in Figure 4 and Figure 5. As can be seen, our method has lower vertex errors in the hidden surface

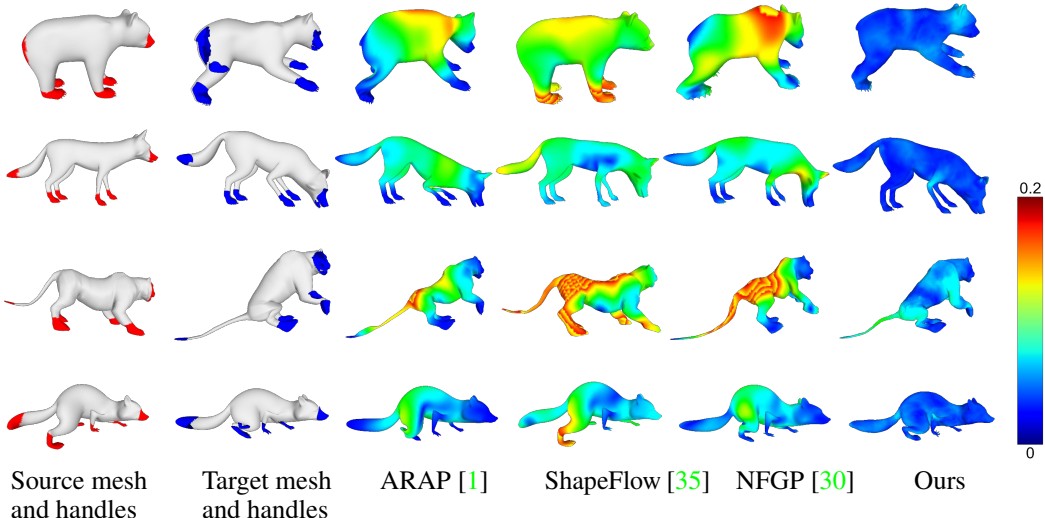

| Source mesh and handles | Target mesh and handles | ARAP [1] | ShapeFlow [35] | NFGP [30] | Ours |

Figure 4: Comparison against ARAP [1], ShapeFlow [35], and NFGP [30] on new motions. We visualize the vertex euclidean distance errors as color maps.

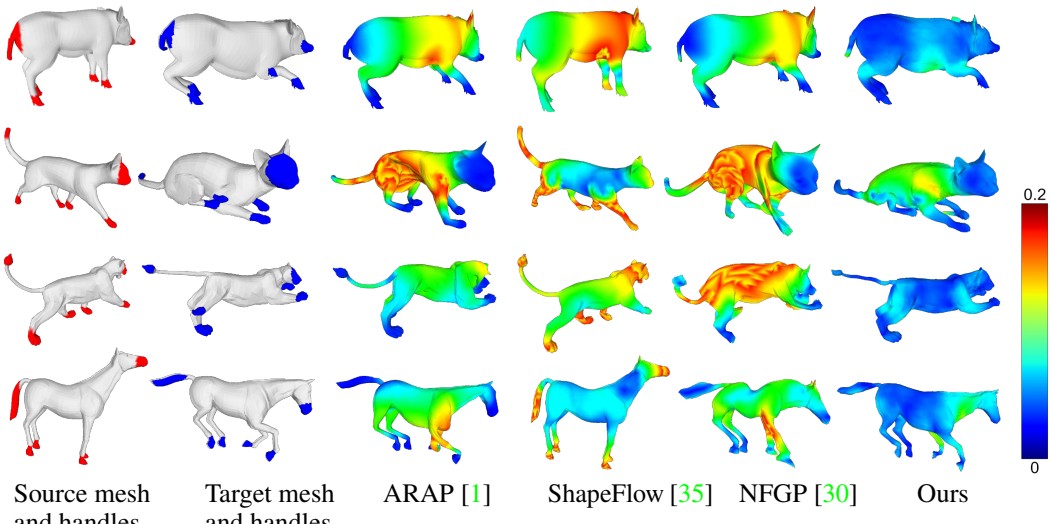

| Source mesh and handles | Target mesh and handles | ARAP [1] | ShapeFlow [35] | NFGP [30] | Ours |

Figure 5: Comparison against ARAP [1], ShapeFlow [35], and NFGP [30] on the S2 test set of DeformingThing4D-Animals and unseen shapes of Deformation Transfer [70]. We visualize the vertex euclidean distance errors as color maps. Our approach generalizes better in comparison to ShapeFlow and NFGP and produces natural looking deformations (in comparison, ARAP generates rubber-like deformations).

regions since we use data-driven deformation priors, instead of employing hand-crafted regularizers to enforce surface smoothness. The generalization ability to unseen deformations is improved by learning deformation fields for local surfaces, instead of modeling global deformations.Compared to ARAP, ShapeFlow, and NFGP, we can produce more realistic results for complicated actions in the 3rd and 4th rows of Figure 4. The deformation results presented in Figure 5 demonstrate that our method can generalize to unseen identities, and is also verified quantitatively in Table 1, where our method consistently outperforms all baselines.

**User-specified handles.** To evaluate the generalization performance of our approach on unseen identities using user-provided handle displacements that are used in interactive editing applications, we use random translations of handles applied to animals from TOSCA [71] as input. As depicted

| Method | New motions (S1) | | | Unseen identities (S2) | | | Deformation Transfer | | |
|---|---|---|---|---|---|---|---|---|---|
| | $\ell_2\downarrow$ | CD$\downarrow$ | FNC$\uparrow$ | $\ell_2\downarrow$ | CD$\downarrow$ | FNC$\uparrow$ | $\ell_2\downarrow$ | CD$\downarrow$ | FNC$\uparrow$ |
| ARAP [1] | 5.568 | 2.312 | 95.35 | 9.794 | 2.308 | 94.89 | 5.145 | 3.475 | 91.21 |
| ShapeFlow [35] | 21.03 | 3.494 | 89.69 | 32.08 | 3.925 | 90.73 | 33.72 | 4.093 | 86.36 |
| NFGP [30] | 11.77 | 3.130 | 93.34 | 15.96 | 3.364 | 91.80 | 18.90 | 4.150 | 82.54 |
| Ours-VDF | 3.590 | 1.887 | 86.01 | 2.368 | 1.837 | 86.99 | 3.111 | 9.164 | 78.63 |
| Ours-global | 2.970 | 1.546 | 93.30 | 2.973 | 1.579 | 94.75 | 2.636 | 8.453 | 84.59 |
| Ours-3D UNet | 1.011 | 1.111 | 96.02 | 1.253 | 1.426 | 96.20 | 4.553 | 2.362 | 88.31 |
| Ours-PointNet++. | 0.886 | 1.055 | 95.47 | 1.231 | 1.364 | 95.37 | 4.898 | 2.564 | 85.87 |
| Ours-w/o atten dec. | 1.184 | 1.210 | 95.64 | 1.227 | 1.417 | 96.16 | 5.252 | 2.772 | 84.95 |
| Ours-w/o cano. | 1.018 | 1.063 | 96.40 | 0.969 | 1.258 | 96.62 | 2.660 | 1.934 | 90.96 |
| Ours-full | **0.752** | **0.948** | **96.59** | **0.795** | **1.241** | **96.68** | **2.495** | **1.877** | **91.40** |

Table 1: Quantitative comparisons on the S1 and S2 test sets of DeformingThing4D [39] and the unseen identities of used in Deformation Transfer [70].

in Figure 6, our approach is able to produce naturally-looking deformation results, and shows its advantages compared to ARAP, ShapeFlow, and NFGP. Note that for this demonstration of user-specified handles there exists no corresponding ground-truth.

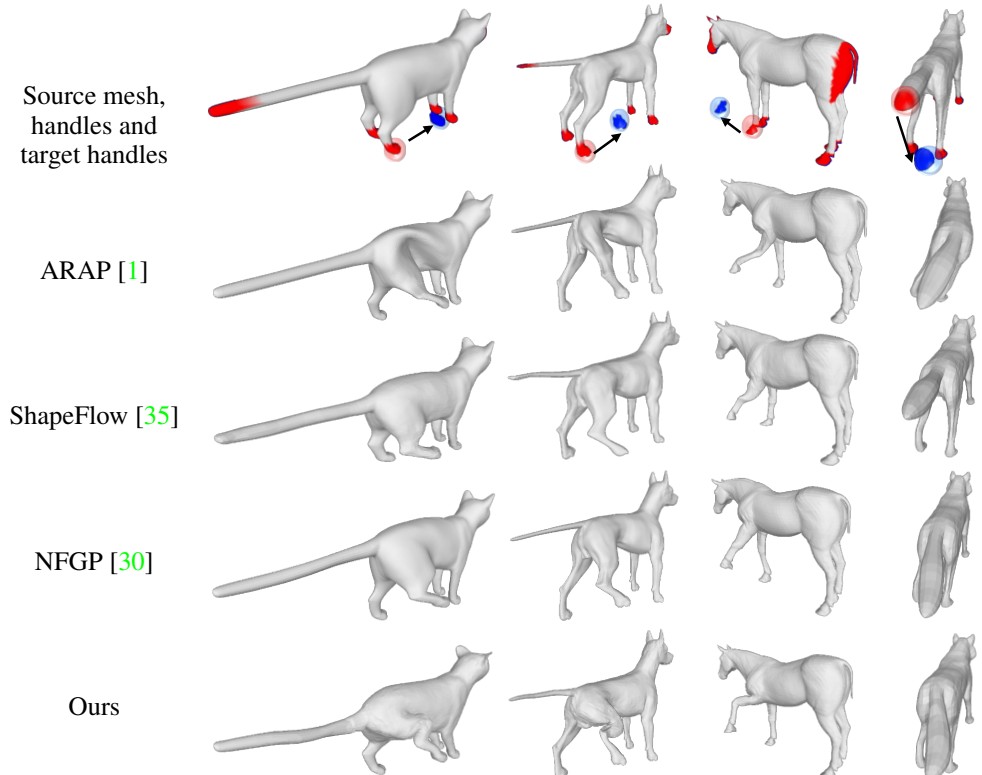

Figure 6: Comparison against ARAP [1], ShapeFlow [35] and NFGP [30] under the setting of user-specified handles on TOSCA dataset [71]. Our method visibly produces the best results.

## 4.2 Ablation Studies

To verify our final model choice, we conducted a series of ablation studies, where we analysed several variants of our deformation fields (see Table 1 and Figure 7).

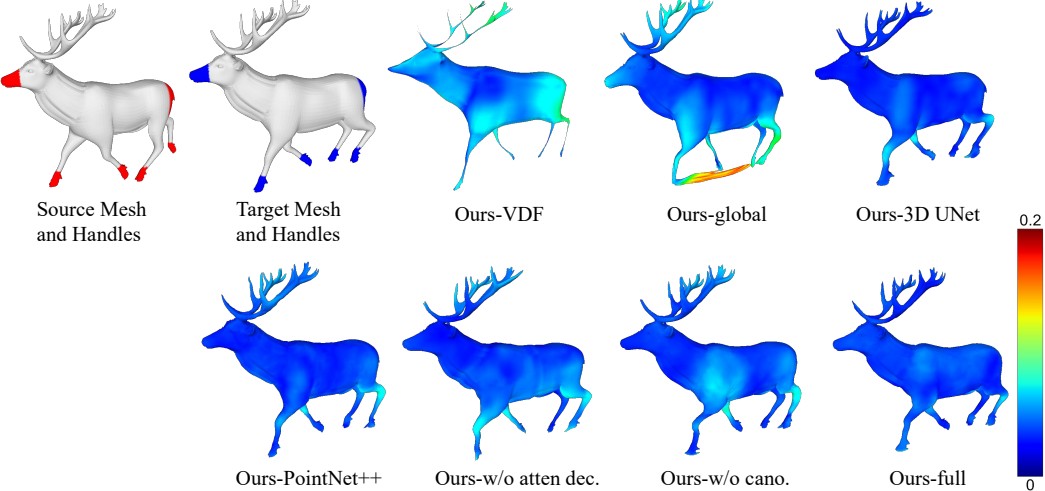

Figure 7: Qualitative ablation studies. Each component of our approach contributes to the final result that has the lowest reconstruction error.

**Volumetric grids vs continuous fields.** As continuous fields are not bound to the resolution of a discrete grid structure, it can better represent complex deformations. The performance degrades when we learn grid-based volumetric deformation fields. This can be seen in the experiment "Ours-VDF" which uses a 3D U-Net [74] to generate volumetric deformation fields of a fixed resolution $64^3$.

**Global vs local deformation fields.** "Ours-global" learns a global continuous field only conditioned on the global latent code. This variant tends to lose detailed information about local surface deformations, and is more difficult to generalize to new motions or identities, leading to inferior results in comparison to our local deformation fields.

**Network Architectures (3D U-Net vs PointNet++ vs Point Transformer).** Compared to grid-based and point-based local deformation descriptors learning, the point transformer-based encoder captures strong global contexts that enforce more global consistency constraints. This provides performance improvements on surface accuracy of deformed meshes. To verify this, we conducted an experiment with "Ours-3D-UNet," which learns a volumetric feature map through a 3D U-Net, and then predicts deformation fields based on queried features via tri-linear interpolation operations. Additionally, we compare with "Ours-PointNet++," which replaces the point transformer encoder with PointNet++ [75].

**With vs without Attention-based feature querying.** The attention-based feature query mechanism can flexibly and effectively select the most relevant deformation descriptors for a query point, resulting in improved performance over feature interpolation purely based on euclidean distances. A deformation decoder that for example uses an interpolation with weights that are purely based on euclidean distance instead ("Ours-w/o atten. dec."), leading to significantly higher errors, particularly in terms of the $\ell_2$ vertex error.

**With vs without canonical poses.** Learning shape deformations via canonicalization improves the generalization to source meshes in different poses. Learning without canonicalization ("Ours-w/o cano."), i.e., learning shape deformations directly between two arbitrary poses, results in considerably higher surface errors.

### 4.3 Intermediate results of canonicalization

In Figure 8, we visualize our intermediate results of canonicalization. As can be seen, our method can project source meshes with arbitrary poses into a canonical space with a same pose.

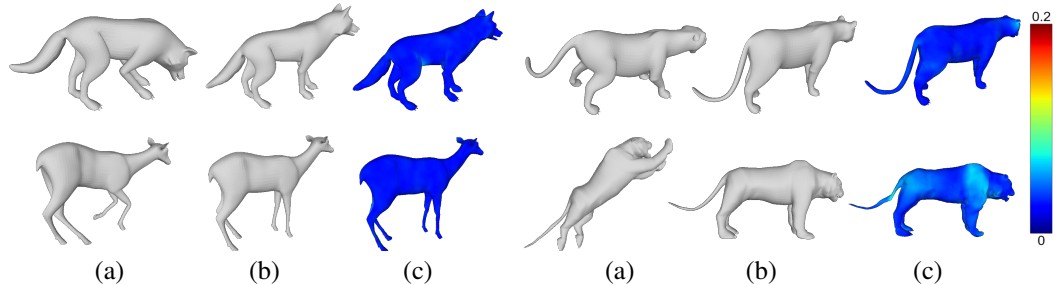

Figure 8: The intermediate results of our canonicalization. (a) Source mesh. (b) Canonical mesh. (c) Our canonicalized mesh.

## 4.4 Limitations

While compelling results have been demonstrated for shape manipulation, a few limitations still exist in our approach that can be addressed in future work. Our approach only needs sparse user input in form of handles which can be moved to create a new deformation state. While this allows for quick editing, a possible extension is to add rotations to the handles. This could be done by leveraging a different deformation representation such as a SE(3) field which is composed of a displacement and a rotation field. Note that our displacement representation is able to represent general deformations, but might require more user handles. Due to the limitations of the DeformingThing4D-Animals [39] dataset in terms of available models and poses, our approach may suffer from the generalization to out-of-distribution models and extreme poses. Additionally, the output of our model, as with other learning-based methods, may be affected by biases in the training dataset that can limit generalization. We believe this issue can be relieved by a larger training dataset and a richer data augmentation strategy in future work. Lastly, our training scheme only considers handles that are selected from a set of candidate parts of the models, thus, limiting the regions the user can interact with. Enriching the candidate handles during training is potentially helpful for allowing free handle placement.

## 5 Conclusion

In this work, we introduced *Neural Shape Deformation Priors*, a novel approach that learns mesh deformations of non-rigid objects from user-provided handles based on the underlying geometric properties of shapes. To enable shape manipulation for source meshes with different poses, we choose to learn shape deformations via canonicalization where the source mesh is first transformed to the canonical space through a backward deformation field and then deformed to the target space through a forward deformation field. For deformation field learning, we propose Transformer-based Deformation Networks (TD-Net) that represent a shape deformation as a composition of local surface deformations. Our experiments and ablation studies demonstrate that our method can be applied to challenging new deformations, outperforming classical optimization-based methods such as ARAP [1] and neural networks-based methods such as ShapeFlow [35] and NFGP [30], while showing a good generalization to previously unseen identities. We see our method as an important step in the development of 3D modeling algorithms and softwares and hope to inspire more research in learning-based shape manipulation.

**Societal impact.** Our work provides an algorithm for natural-looking shape editing, which can simplify tedious procedures in 3D content creation and empower artists in the movie and game industries. It further has the potential to enrich 3D data with additional deformed shapes, and could thus help improve the performance of other practical application techniques that rely on large quantities of 3D ground-truth for training. Yet, misuse of our shape manipulation algorithm could enable fraud or offensive content generation.

**Acknowledgement.** This work is supported by a TUM-IAS Rudolf Mößbauer Fellowship, the ERC Starting Grant Scan2CAD (804724), and Sony Semiconductor Solutions Corporation. We would also like to thank Angela Dai for the video voice over.

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
