# OpenReview forum: "Neural Shape Deformation Priors"
_NeurIPS.cc/2022/Conference — NeurIPS 2022 Accept_

### Official Review · Reviewer_uKtN · 2022-07-11

**Rating:** 6
**Confidence:** 4
**Soundness:** 3 good
**Presentation:** 3 good
**Contribution:** 3 good

**Summary:**

In this paper is introduced an approach to learn mesh deformations of dynamic bodies from user-provided handles. To manipulate the source meshes into different poses, the shape deformations are learned via canonicalization, i.e., the source mesh is first represented in the canonical space, and after that, that representation is transformed to the target space. To this end, two backward and forward deformation fields are considered, that are learned by transformer deformation networks. Both quantitative and qualitative analysis are provided, as well as a comparison with two competing methods, showing promising results. Additional ablation studies are also included in the work.

**Questions:**

Please, see the strengths and weaknesses section for questions and suggestions.

**Limitations:**

The authors have addressed the limitations properly.

**Strengths And Weaknesses:**

In general, the paper is clear enough, it is well written, and all the technical details are given in the document. Motivation and related work are concise, and the authors clearly show their contribution. To be honest, I do not have many issues with this submission in its current form. Next, I have some comments and questions.

The deformation priors are learned by the use of a canonical space, i.e., the shape transformation between two arbitrary poses is divided into two steps: a backward deformation to align the source mesh to the canonical space, and a forward deformation to map the canonical space in the target deformation space. While this process increases the complexity of the neural model, as two different models need to be learned, in practice the process is strongly simplified. This idea is simple yet effective. Once the transformation models are learned, a simple 3D displacement, via a deformation field, is considered to deform the mesh.

Even trivial, all operators in Eq. (1) should be defined.

For training, the method needs dense correspondences for the three deformation states (canonical, target, and source). While this can be an easy task for synthetic scenarios and CAD-based models, this could be a very hard task in real scenarios, where noisy point clouds could be considered with a variation of points in the representation.

The authors claim they divide the test set in 143 sequences for seen train identities, and 55 sequences for unseen ones.  In my opinion, this should be clarified in the paper. How was that done? Note that some unseen identities could be the result of a simple linear combination of seen ones, i.e., the unseen motions are really seen ones. I would like to know as the authors can guarantee that division with no additional analysis.

Regarding Figure 4. The authors claim their solution produces visibly the best results. I disagree with that. That conclusion is not actually easy, after checking the corresponding image. Maybe the authors could include the corresponding ground truth mesh, or color-based representation where every color displays a different error.

Observing the rest of results, the proposed method outperforms, in both quantitatively and qualitatively, the competing approaches.

It is worth noting that the user-specified handles could be non-realistic or non-physical-aware. In that case, my question is: could the proposed method obtain the deformation? This could help us to interpret a bit more the learned deformation priors. My doubt is the deformation priors could be just an algebraic representation with no meaning. Could the authors help me with this question?

More extreme poses could be considered as well as some realistic animal meshes (for instance, capturing the full mesh by a real vision sensor that means the mesh includes noisy and partial observations).

---

> ### Author Response · Authors · 2022-08-02
> **Response to Reviewer uKtN**
>
> Q1: Require dense correspondences of CAD-based models.
>
> While our current method uses a dataset where dense correspondences between temporal mesh frames are available, our framework can also be trained on datasets without dense correspondences by some adjustments on inputs and loss functions. Concretely, we can change our method to receive sparse handle correspondences as inputs, and utilize Chamfer distance as the loss function that does not require ground-truth meshes with dense correspondences. In Figure 9 of the revised supplementary material, we visualize test results of such a modified framework. As seen, without dense correspondences for training, our method can still obtain accurate deformations.
>
> Q2: Train/test split.
>
> The train/test split is based on the provided identity and motion names of deforming sequences. We first divide the animations of the dataset into two parts, seen identities and unseen identities. For the animations of seen identities, we further divide it into seen motions of seen identities (used as training set), and unseen motions of seen identities (used as the test set of S1). For animations of unseen identities, we remove those animations whose motions have already appeared in the training set. This way, we guarantee that the motions of unseen identities are not seen during training. Please also refer to Section D in the revised supplementary material for details.
>
> Q3: Provide ground-truth meshes in Fig. 4.
>
> There is no ground-truth mesh in the experiment of the user-provided handles. Thus, one cannot calculate the vertex errors of the deformed meshes. However, our method can still obtain more realistic deformations, such as the leg movement of the deer in the last column.
>
> Q4: Non-realistic or non-physical-aware user-specified handles.
>
> Our method will find the closest deformation of animals that can best explain the provided user handles. Further, our goal of data-driven deformation priors is to obtain deformations that are as realistic as possible. However, our method could be easily trained on non-realistic or non-physical-aware samples and learn the respective deformation behavior. We will clarify this in the final revision.
>
> Q5: Robustness to noisy and partial observations of source mesh.
>
> We directly evaluate our model on noisy and incomplete meshes without fine-tuning. The quantitative results are provided in Tables 3 and 4 of the revised supplementary material. As seen, there are no significant numerical variations between different noise levels and incompleteness ratios. This clearly demonstrates the robustness of our approach to noisy and/or incomplete source meshes.
>
> Q6: Real test on animal scans.
>
> We are happy to evaluate our method on real scans; however, there are not that many datasets that contain real animal scans. We have sent emails to the authors for the dataset access, and are currently waiting for a reply. In parallel, we plan to capture animal scans by ourselves and include the evaluation results in the revised paper. In addition, we evaluate our pre-trained model on the reconstructed animals from real RGB images using the BARC method. As shown in the Figure 8 of the revised supplementary material, our method estimates realistic deformations for reconstructed animals from natural images. This also demonstrates the generalization ability of our method.

---

> > ### Comment · Reviewer_uKtN · 2022-08-09
> > **Response to the authors**
> >
> > Thank you very much for the response. In my opinion, I think the authors have addressed my doubts and comments as well as the those of my colleagues. Maybe, the answer for the section "Non-realistic or non-physical-aware user-specified handles" is not clear enough, but anyway, the rest of the things are correct. Regarding that comment, I cannot understand how the deformation to be obtained is as realistic as possible, due to the fact that we do not have a clear explanation of the model in terms of interpretability. On balance, my rating remains.

---

> > > ### Author Response · Authors · 2022-08-09
> > > **Response to Reviewer uKtN**
> > >
> > > Thanks for the positive feedback!
> > >
> > > Our model learns deformation priors from a dataset containing realistic non-rigid motions. When it is directly evaluated on non-realistic or non-physical-aware handles, it will try to find the most similar realistic deformation that can best explain the given handles.
> > > However, we can easily transfer the ideas to non-realistic or non-physical priors by using an appropriate dataset.
> > > We are happy to discuss this in more detail and include an example experiment in our camera-ready version.
> > >
> > > Regarding interpretability, we agree that this is not non-trivial; however, we can analyze and precisely evaluate the output of our model; e.g., as shown in our experiments, our method predicts more realistic deformations than state-of-the-art baselines such as ARAP, NFGP, and ShapeFlow. We will further add a clear explanation of the model in terms of interpretability.

---

### Official Review · Reviewer_oxNS · 2022-07-12

**Rating:** 5
**Confidence:** 3
**Soundness:** 3 good
**Presentation:** 3 good
**Contribution:** 3 good

**Summary:**

This paper presents a neural deformation method, which utilize the deformation priors in a large-scale dataset. This method predicts a continuous deformation field in space. The input model can be of any pose. The proposed method first deforms the input model back to the canonical space, and then deforms it to the target that satisfies the constraints given by the user. The authors propose Transformer-based Deformation Networks (TD-Nets), which learns encoder-based local deformation fields on point cloud approximations of the input mesh and outputs the deformation.

**Questions:**

-- What is the difference between the randomly sampled surface point cloud P_S and the querying spatial points Q_S? More description about the querying point set should be given.
Also in Line 195, ‘querying spatial points (P_S, P_C, P_T)’ should be ‘querying spatial points (Q_S, Q_C, Q_T)’.
--Are equations (6),(7),(8) all used during training, or equations (6) and (7) just for the derivation of equation (8)?
--There are many modules in the proposed network, PAB, PTB, Point transformer encoder, Attentive deformation decoder, and also many features and local codes denoted as Z. I am wondering how the data goes through these modules. It would be better to have a network diagram showing how the modules are combined.


**Limitations:**

The authors have discussed the limitations.

**Strengths And Weaknesses:**

Strengths
--The method is technically sound.
--The proposed method is carefully validated and has been compared to the representative deformation method ARAP and a neural-based method NFGP.

Weaknesses
-- The description needs to be improved.

---

> ### Author Response · Authors · 2022-08-02
> **Response to Reviewer oxNS**
>
> Q1: Difference between $\mathcal{P_S}$ and $\mathcal{Q_S}$.
>
> $\mathcal{P_S}$ is the sampled point cloud from the surface of source mesh $\mathcal{S}$. In contrast, $\mathcal{Q_S}$ is the sampled non-surface point set from the 3D space of source mesh $\mathcal{S}$. We obtain $\mathcal{Q_S}$ by adding gaussian noise permutations along the normal directions of $\mathcal{P_S}$.
> Please refer to Section D in the revised supplementary material for the detailed description about $\mathcal{P_S}$ and $\mathcal{Q_S}$.
>
> Q2: L195.
>
> Thank you for pointing this out. The querying non-surface point sets in 3D space should be denoted as $(\mathcal{Q}_\mathcal{S}, \mathcal{Q}_\mathcal{C}, \mathcal{Q}_\mathcal{T})$. We fixed it in the revised paper.
>
>
> Q3: Loss functions in Equation (6), (7), (8).
>
> Our model consists of two training stages. In the first stage, the backward and forward deformation networks are individually trained using the loss functions defined in Equation (6) or (7), respectively. In the second stage of end-to-end training, the whole network is trained with the loss function defined in Equation (8). Please also refer to lines 214--221 for further implementation details.
>
>
> Q4: More details of network architectures.
>
>
> Please refer to the details described in Section A in the supplementary material.

---

> > ### Comment · Reviewer_oxNS · 2022-08-08
> > **Response to the authors**
> >
> > The authors solve my problem and also answer other reviewers' questions. So I keep the attitude of acceptance.

---

### Official Review · Reviewer_ggj2 · 2022-07-12

**Rating:** 4
**Confidence:** 3
**Soundness:** 2 fair
**Presentation:** 2 fair
**Contribution:** 2 fair

**Summary:**

This paper deals with the problem of surface deformation by training a Transformer network on point cloud. The key idea is, they adopt a canonical model or space as often used in human modeling and design a backward and forward deformation network to deform from any source model to target model.

**Questions:**

I have some questions or confuse about the experiments:
1) In Tab.1, the L2 error of the ARAP method increased quite a bit on unseen identities. Why would happen? ARAP is not a learning method, so it shouldn't get affected whether they have seen those identities before.
2) For the learning based baseline method, NFGP, I'm wondering whether it is re-trained under this DeformingThing4D-Animals dataset.
3) What is the major technical contribution the authors want to claim? The point transformer?

**Limitations:**

The authors have mentioned their limitations in the future work which is good, but those mentioned limitions are fundamental problems that have to be dealt with. It would be a bit better to demonstrate results on general models that are not from the DeformingThing4D-Animals dataset.

**Strengths And Weaknesses:**

Strengths:
First, solving the mesh deformation under the movement of some user-specified handles is a typical problem in graphics and the proposed idea is straightfoward and pretty easy to understand. So the paper is well written and easy to follow. They have demonstrated better performance on the DeformingThing4D-Animals dataset.

Weakness:
They only demonstrated the performance on the DeformingThing4D-Animals dataset. Although they have tested on unseen identities, the models in the dataset are quite similar. But the mesh deformation itself is a general problem and we shouldn't assume the models are all animals. From this perspective, the generalization ability is a big issue for the proposed method. On the other hand, the compared methods are more general and can directly apply to various kinds of models. Therefore, the comparison is just not fair. From my understanding, the network is overfitted to this dataset.
The rotation is not included, instead the network only predicts the displacement of vertices. This could be a big issue for surface deformation. The authors have mentioned this in the future work, but as a fundemental problem, I think it should have first priority to be tackled.

---

> ### Author Response · Authors · 2022-08-02
> **Response to Reviewer ggj2**
>
> Q1: Technical Contribution.
>
> The main technical contribution is the transformer-based deformation network that represents a shape deformation as a composition of local surface deformations. This allows us to learn non-linear, localized deformations in a data-driven way based on learned features of the underlying shape geometry. In contrast to global deformation models like ShapeFlow, our local deformation model enables significantly better generalization ability to unseen motions.
>
> Q2: The generalization issue.
>
> We wish to clarify that our goal is to learn deformation priors of a specific class of objects (e.g. quadruped animals) and not a generalized deformation model without a class prior. We add an additional baseline of ShapeFlow that is also specific to a class. Please refer to Table 5 and Figures 6 and 7 in the revised supplementary material. From the result analysis, we see that our method can learn more accurate deformation priors compared to ShapeFlow.
>
>
> Q3: Not learning rotations.
>
> In general, displacements are able to represent arbitrary deformations. However, we agree that SE(3) fields could be a more efficient representation of deformations and can potentially lead to higher quality (given a fixed spatial resolution). We will add an in-depth discussion about our design choice.
>
>
> Q4: Higher L2 error of ARAP on unseen identites.
>
> We wish to clarify that the deformations of unseen identities are more complicated than those of unseen motions, which can cause the higher prediction errors of ARAP.
>
>
> Q5: The training of NFGP.
>
> NFGP is a deep optimization-based method and cannot learn general deformation priors of a dataset. It overfits the neural network to each provided input during inference.

---

### Official Review · Reviewer_KXsR · 2022-07-14

**Rating:** 5
**Confidence:** 3
**Soundness:** 3 good
**Presentation:** 3 good
**Contribution:** 2 fair

**Summary:**

This paper proposes a deep-learning framework to model shape deformations, especially in the context of modifying a surface with user-defined input locations shifted to their desired locations. The motivation is that prominent methods like ARAP are restrictive in their results, partly due to pure geometric action, and that learning these deformations from data allows for more richer semantic variability in the deformations.

The authors make use of a transformer-based architecture and implement a two-staged deformation learning paradigm. Specifically, a backward deformation network deforms the shape into a canonical position. A forward deformation network, inputs this canonicalized shape, and the desired target handle information to output a deformed mesh. The training is done in two stages, with three different loss functions in action accordingly.

The authors demonstrate their method on the DeformingThing4D-Animals dataset and show favorable results both numerically and visually in comparison to competing prior works especially ARAP and NFGP.


**Questions:**

Questions

- In the limitations, what do you mean by model rotation? like equivariance to rotations?
- Is there a reason why you achieve very decent smooth results, especially for the canonical pose visualizations in the supplementary, despite no explicit imposition of some regularity?
- How exactly do you get ground truth matching between Qs and Qt? As far as I understand Qs, and Qt are the points sampled in space and in the vicinity of the shape (and not on it), how do you then establish correspondence between points in 3D space? when I presume: only surface-to-surface matching is available?


**Ethics Review Area:**

["I don’t know"]

**Limitations:**

See Weaknesses. Overall, I found this paper interesting and well compiled and I am inclined to weigh in positively as a pre-discussion rating. However, there are some outstanding issues that need clarification and will wait for the discussion phase to make a more informed opinion.

**Strengths And Weaknesses:**

Strengths

- Overall, I found this to be an interesting paper, with good motivation, well-written and well-compiled experiments, especially in the supplementary
- Clever use of the transformer system applied to modifying shapes with target handle information for shape editing
- In addition, I found the limitation section to be honest and well explained


Weaknesses

- The notation is a bit cumbersome and hard to keep up with. Although I did catch up after multiple reads, section 3 could be written more clearly. I specifically suggest having a clear depiction for so many notations on various point clouds -   C, O, Q, P, and T
- As rightly pointed out in the paper, the method is essentially supervised requiring dense correspondences throughout. Although not a deal-breaker this must be noted as a drawback since availability of such data is scarce.
- I found little attempt to combine the proposed learning-based approach with the optimization approaches like ARAP. Could there be a combination of these losses and does that lead to better generalization (and possibly fewer data requirements?)
- From an evaluation perspective, this paper is very restrictive in its demonstration and the results are only shown for 1 dataset. As a result, the transfer learning capabilities (for e.g. to 4-legged animals in TOSCA)  are largely unknown.
- Although there is a short commentary on robustness, very little is demonstrated apart from resampling the meshes.
- Please do contrast, explain, and/or compare with a relevant recent work: Jiang, Chiyu, et al. "Shapeflow: Learnable deformation flows among 3d shapes." Advances in Neural Information Processing Systems 33 (2020): 9745-9757.
- Typo: Line 195, is that (Qs, Qc, and Qt)?

---

> ### Author Response · Authors · 2022-08-02
> **Response to Reviewer KXsR**
>
> Q1: Comparisons on the dataset of Deformation Transfer and TOSCA.
>
> As suggested, we directly evaluate our pre-trained model on other animal datasets by providing additional quantitative results on the dataset used in Deformation Transfer. TOSCA does not have correspondences between different poses of the same animal, and hence does not easily provide handle displacements as input. Thus, we provide the comparison under the setting of using user-specified handles as inputs.
> Please refer to Table 5 and Figure 6 in the revised supplementary material. We can observe that our method generalizes well to the other datasets without the need of re-training our model.
>
>
> Q2: Discussion and comparison against ShapeFlow.
>
> In lines 109--112 of the revised paper, we detail the difference and connection between ShapeFlow and our method. We also include a ShapeFlow comparison. Please refer to the Table 5 and Figures 6 and 7 in the revised supplementary material. Our method can predict more accurate deformations both quantitatively and qualitatively.
>
> Q3: Robustness to noisy source meshes.
>
> We directly evaluate our model on noisy meshes without finetuning. The quantitative results are provided in Table 3 of the revised supplementary material. With the noise becoming larger, the performance of our method experiences only slight variation; however, this demonstrates the robustness of our method to noisy source meshes.
>
> Q4: Robustness to incomplete source meshes.
>
> We directly evaluate our model on incomplete meshes without finetuning. The quantitative results are provided in Table 4 of the supplementary material. As seen, there are no significant numerical variations between different  incompleteness ratios. This clearly demonstrates the robustness of our approach to incomplete source meshes.
>
> Q5: Generalization to real scans.
>
> We are happy to evaluate our method on real scans; however, there are not that many datasets that contain real animal scans. We have sent emails to the authors for the dataset access, and are currently waiting for a reply. In parallel, we plan to capture animal scans by ourselves and include the evaluation results in the revised paper. In addition, we evaluate our pre-trained model on the reconstructed animals from real RGB images using the BARC method. As shown in the Figure 8 of the revised supplementary material, our method estimates realistic deformations for reconstructed animals from natural images. This also demonstrates the generalization ability of our method.
>
> Q6: Combination of ARAP and our method.
>
> We agree that a combination of ARAP and our method might reduce the amount of required data for training. However, at the same time, this would mitigate the advantage of our data-driven method which facilitates the learning of non-linear, localized deformation properties based on features of the underlying shape geometry. We will be happy to include an experiment to illustrate the trade-off between "more ARAP for regularization" vs "pure data-driven learning".
>
> Q7: Limitation of requiring dense correspondences as supervision.
>
> While our current method uses an existing dataset where dense correspondences between temporal mesh frames are available, our framework can also be trained on datasets without dense correspondences by some adjustments on inputs and loss functions. Concretely, we can change our method to receive sparse handle correspondences as inputs, and utilize Chamfer distance as the loss function that does not require ground-truth meshes with dense correspondences. In Figure 9 of the revised supplementary material, we visualize test results of such a modified framework. As can be seen, without dense correspondences for training, our method can still obtain accurate deformations.
>
>
> Q8: Notations.
>
> We provide a summary in Table 1 of the revised supplementary material to clearly define the notations throughout the paper.
>
>
> Q9: L195.
>
> Thank you for pointing out the issue. The querying of the non-surface point sets in the 3D space should be denoted as $(\mathcal{Q}_\mathcal{S}, \mathcal{Q}_\mathcal{C}, \mathcal{Q}_\mathcal{T})$. We have fixed it in the revised paper.
>
>
> Q10: Rotation Fields in the limitation.
>
> We plan to further decompose a deformation field as a rotation field and translation field in the future. Please refer to lines 275--280 in the revised main paper for a clearer description.
>
>
> Q11: Smooth canonicalization visualization.
>
> The reason is that we learn continuous deformation fields defined in 3D space, thus enabling smooth mesh deformations.
>
> Q12: Obtain the ground-truth matching between $\mathcal{Q_S}$, $\mathcal{Q_C}$, and $\mathcal{Q_T}$.
>
> The ground-truth matching between non-surface spatial point set of  $\mathcal{Q_S}$, $\mathcal{Q_C}$, and $\mathcal{Q_T}$ is based on the dense correspondences between surface meshes. Please refer to Section D in the revised supplementary material for the data-preprocessing details.

---

### Author Response · Authors · 2022-08-02
**To Reviewers and ACs**

We thank the reviewers for the constructive comments. It is very encouraging to see that the reviewers found our paper "well written" (R1, R2, R4), "interesting" (R1), well motivated (R1, R4), clear (R2, R4), our problem relevant (R2), our method "technically sound" as well as "carefully validated" (R3). In the following, we address the reviewer comments. These responses, together with those for minor issues, have already been included in the revised paper (see updated version).

---

### Meta-Review · Area_Chair_MTFX · 2022-08-23

**Recommendation:** Accept
**Confidence:** Certain

**Metareview:**

While some of the scores on this paper are mixed, even the negative reviews highlight the quality and interest of the work and have specific (and somewhat debatable) technical concerns.  Overall, the AE recommends accept, especially in light of the detailed and thoughtful responses during the rebuttal phase.

In the camera ready, the authors are encouraged to see if they can squeeze some of the new results (e.g., transfer learning attempt in Figure 6 and comparisons to Shapeflow) in the main body of the paper, where they're more likely to be noticed.

**Award:**

No

---

### Decision · Program_Chairs · 2022-09-14

Accept